# Satellite-Based Carbon Estimation in Scotland: AGB and SOC

Chun Ki Chan *, Carla Arus Gomez, Anish Kothikar and P. M. Baiz-Villafranca *

Department of Computing, Imperial College, London SW7 2BX, UK
* Correspondence: chun.k.chan16@alumni.imperial.ac.uk (C.K.C.); p.m.baiz@imperial.ac.uk (P.M.B.-V.)

**Abstract:** The majority of state-of-the-art research employs remote sensing on AGB (Above Ground Biomass) and SOC (Soil Organic Carbon) separately, although some studies indicate a positive correlation between the two. We intend to combine the two domains in our research to improve state-of-the-art total carbon estimation. We begin by establishing a baseline model in our study area in Scotland, using state-of-the-art methodologies in the SOC and AGB domains. The effects of feature engineering techniques such as variance inflation factor and feature selection on machine learning models are then investigated. This is extended by combining predictor variables from the two domains. Finally, we leverage the possible correlation between AGB and SOC to establish a relationship between the two and propose novel models in an attempt to outperform the state-of-the-art results. We compared three machine learning techniques, boosted regression tree, random forest, and xgboost. These techniques have been demonstrated to be the most effective in both domains. This research makes three contributions: (i) Including Digital Elevation Map (DEM) as a predictor variable in the AGB model improves the model result by 13.5 % on average across the three machine learning techniques experimented, implying that DEM should be considered for AGB estimation as well, despite the fact that it has previously been used exclusively for SOC estimation. (ii) Using SOC and SOC Density improves the prediction of the AGB model by a significant 14.2% on average compared to the state-of-the-art baseline (When comparing the R2 value across all three modeling techniques in Model B and Model H, there is an increase from 0.5016 to 0.5604 for BRT, 0.4958 to 0.5925 for RF and 0.5161 to 0.5750 for XGB), which strengthens our experiment results and suggests a future research direction of combining AGB and SOC as a joint study domain. (iii) Including AGB as a predictor variable for SOC improves model performance for Random Forest, but reduced performance for Boosted Regression tree and XG Boost, indicating that the results are specific to ML models and more research is required on the feature space and modeling techniques. Additionally, we propose a method for estimating total carbon using data from Sentinel 1, Sentinel 2, Landsat 8, Digital Elevation, and the Forest Inventory.

**Keywords:** terrestrial carbon estimation; total carbon estimation; soil organic carbon; above ground biomass; machine learning; satellite imagery

## 1. Introduction

In remote sensing, AGB and SOC predictions are usually seen as two separate problems; however, there have long been research papers suggesting in order to understand the effect of climate change and the total carbon sink on earth, we need to study land as a terrestrial system. Early efforts provide carbon estimates in the vegetation and soils in Great Britain [1] based on land cover and allometric equations. Although the accuracy is limited and the carbon map produced is solely based on land cover estimations, the effort shows the research interest in providing information on AGB and SOC together. Scurlock and Hall [2] looked at the problem from a grassland perspective and introduced the concept of "missing sink", which refers to the natural carbon sink that we have not been able to recognize before. The idea of grassland carbon sink is later extended [3], SOC, AGB, Grazing and Climate Change are associated together and have a proven strong relationship

with one another. Carlos et al. [4] looked into the tropical forest landscape as an ecosystem to better understand the global carbon cycle and total carbon stock.

Total Carbon Stock has also been analyzed on the scale of a terrestrial ecosystem. Sothe et al. [5] analyzed the total carbon stock in Canada by using different research sources. Although most sources either focus on SOC and AGB separately, it is clear that both are required to provide useful information that can affect government-level decision-making and possibly the voluntary carbon market. Despite the abundance of separate research on SOC and AGB, there is a need to better understand total carbon through joint research on SOC and AGB. The reason behind this is that terrestrial ecosystems are intertwined between AGB and SOC. The dynamics of carbon in terrestrial ecosystems are determined by processes such as respiration, combustion and decomposition [6].

Previous literature on total carbon estimation is summarised in Table 1. Analysis in this domain is predominantly done using regression analysis [7–10]. Others [11] used ML techniques suggested in the AGB domain and SOC domain, but only random forest is explored. This research mainly focuses on grassland and forest ecosystems. While some of them used elevation data, none used satellite data from the Sentinel family (Sentinel 1, 2, 3) and LandSat 8 nor did any use SOC as a predictor. Some found a positive correlation between AGB and SOC [7,9] but those only used field samples and LiDAR data which is not scalable to a larger study region.

**Table 1.** Summary of joint research on AGB/SOC estimation.

| Literature | Description | ML Techniques | Vegetation Cover | Data Sources | Region/ Year of Study | Use Digital Elevation | Use AGB as a Predictor of SOC | Use SOC as a Predictor of AGB |
|---|---|---|---|---|---|---|---|---|
| Gebeyehu et al. [7] | Studied the relationship between AGB/SOC and concluded from the regression analysis that the significant positive correlation suggests AGB as a useful predictor of SOC. | LR | Forest Ecosystem | Global Wood Density database, field samples | Awi Zone, northwestern Ethiopia, 2019 | No | Yes | No |
| Wang et al. [11] | Created multivariate RF model to estimate topsoil SOC and AGB, using meteorological factors, Satellite images and Digital Elevation. Discovered a strong positive correlation between AGB and SOC in desert steppe and the steppe desert of rocky mountains. Provided evidence that AGB and air temperature should be given more attention in SOC prediction. | RF ($R^2$ = 0.62, RMSE = 89.37 for AGB and $R^2$ = 0.72, RMSE = 3.99) | Grassland | MODIS, LandSat 5, ASTER (Elevation) | Loess Plateau, China, 2017 | Yes | No | No |
| Vicharnakorn et al. [8] | First, perform land classification, then developed an AGB estimation model from field samples and Landsat Thematic Mapper (TM) image. Various bands were analyzed with multiple regression analysis to study the correlation between AGB and RS bands. This is later put together with field-measured SOC to present a total carbon estimation for the study area | LR; Correlation in regression model between AGB and RVI/SAVI/SR is 0.931 | Forest Ecosystem | Landsat Thematic Mapper (TM) image | Savannakhet Province, Lao People's Democratic Republic, 2014 | No | No | No |
| Rasel [9] | Analysed AGB, elevation, bulk density and soil PH in the context of SOC. Found a positive correlation between SOC and elevation and AGB | Linear Regression for AGB estimation, which is then used to study SOC content. Correlation of 0.79 between AGB/SOC and 0.84 between AGB/Elevation | Forest Ecosystem | LiDAR, DEM, AGB | Chitwan district, Nepal, 2013 | Yes | Yes | No |
| Yang et al. [10] | Examined the relationship between AGB/SOCD and found a strong positive correlation, suggesting plant production largely determines SOC content in alpine grassland. EVI derived from MODIS also has a strong correlation between AGB and SOC Density and is treated as a predictor variable for SOC estimation. | Regression Analysis ($R^2$ SOCD/AGB = 0.39) | Alpine Grassland | MODIS-EVI | Qinghai-Tibetan Plateau, China, 2008 | No | No | No |
| Scurlock & Hall [2] | Discovered that grassland and savannas contribute to more 'missing sink' than previously anticipated, suggesting possible future research directions | N/A | Grassland | Field measurements and Previous studies | Global, 1997 | N/A | N/A | N/A |
| Milne & Brown [1] | Created total carbon map for Great Britain by combining previous studies on biomass partitioning, census of forests, ecological surveys of sample areas and RS land cover map. Suggesting early interest in the total carbon estimation domain combining SOC/AGB | N/A | General to the UK | Past studies | Great Britain, 1995 | N/A | N/A | N/A |

State-of-the-art models are used as a baseline for comparison. The SOC baseline is trained on Sentinel 1A, Sentinel 2A (Including vegetation indices) and Digital Elevation Data. This is based on previous studies: Zhou et al. [12] used Sentinel 1/2 and DEM data, whereas Emadi et al. [13] used Digital Elevation Data, Gholizadeh et al. [14] used Sentinel

2 and Digital Elevation data. On the other hand, the AGB baseline model is based on Li et al. [15] which used LandSat 8 and Forest Inventory data as they contain information on multiple wavelengths and woodland classification which are essential to above-ground biomass estimates.

The experimentation involves a number of stages. To begin, we apply state-of-the-art models/input variables to our site in order to create a baseline model from which we can improve. The baseline models are then improved through feature engineering and an examination of the relationships between AGB and SOC. Although there has been research on the correlation between AGB and SOC, none has examined the possibility of using one as an input to predict the other. As a result, our investigation in AGB and SOC has two primary objectives:

1. Mix inputs that were previously thought to be useful only for AGB or SOC. For instance, given that digital elevation is known to be a good predictor for SOC [12], we ask whether it also provides insight into AGB prediction.
2. Since there exists a positive correlation between AGB and SOC [9], we attempt to improve existing models by including SOC/SOC Density to predict AGB and using AGB to predict SOC.

## 2. Materials and Methods

### 2.1. Study Area

The study area covers a part of a rural area in Scotland, United Kingdom. It is located east of Glasglow and south of Edinburgh between (Latitude, Longitude) = (55.754194°, −3.703772°) NW and (55.4075244°, −2.7696528°) SE. The study area is shown in Figure 1, it is covered by a mix of forests, grassland and other urban areas. From the National Forest Inventory Woodland Scotland data [16], the identified forest inventory is mainly covered by woodland (92.91%), mixing some grassland (2.62%) and urban areas (0.20%).

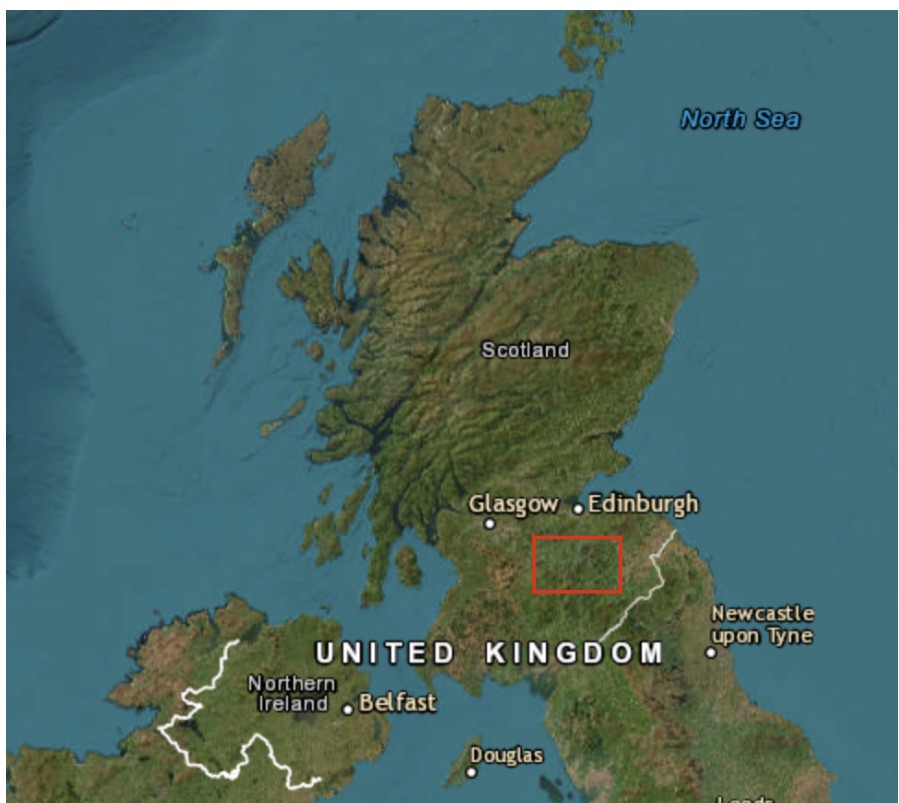

**Figure 1.** USGS Earth Explorer: Scotland, United Kingdom—A rural area south of Edinburgh.

The climate in this region is classified as temperate oceanic, with mild temperatures and high levels of precipitation throughout the year. The average maximum air temperature for Scotland between 1981 and 2010 was 10.7 °C, while the average minimum air temperature was 4.2 °C [17]. The topology and landforms of southern Scotland are the results of complex geological processes that occurred over millions of years. The region was once covered by the Iapetus ocean about 500 million years ago. The collision of two tectonic plates caused the ocean to close, leading to the formation of the Caledonian Orogeny, which formed the roots of the Scottish Highlands, Lake District, and North Wales. The closure of the Iapetus took 80 million years, leaving behind sedimentary rocks visible today, such as the accretionary prism, which is a mixture of mud, sand, and shales. The Southern Uplands, bounded by the South Uplands fault in the north and the Iapetus suture in the south, are a visible representation of the accretionary prism. The geological map of the area shows different rock types, such as greywackes, shales, and igneous emplacements, and fault lines that thrust the Southern Uplands onto land. These rocks exhibit the power of plate tectonics that is still present in the area [18].

*2.2. Data Sources*

2.2.1. Soil Organic Carbon

A total of 25.021 pre-processed data points representing Soil Organic Carbon at 0–30 cm depth are obtained from Soil Grids 2.0 [19] (2021). Soil Grids 2.0 maps soil properties globally at a resolution of 250 m, taking as input field soil samples from about 240,000 locations worldwide. Soil samples in Soil Grids 2.0 are obtained from ISRIC World Soil Information Service (WoSIS), which provides globally standardized soil profile data [20].

2.2.2. Above Ground Biomass

AGB data is obtained from the Global Above and Below Ground Biomass carbon density map [21] (2020). The dataset [22] is open-sourced by NASA ORNL (Oak Ridge National Laboratory, Oak Ridge, TN, USA) featuring AGB at a resolution of 300 m. The global map is compiled from published literature using a harmonization approach, matching maps of tundra, grassland and annual crops.

*2.3. Predictor Variables*

The predictor variables used in this paper are Sentinel 1 [23], Sentinel 2 [24], Landsat 8 [25], DEM derivatives [26] and Scotland Forest Inventory Data [16]. These variables are obtained from various sources and converted into raster data (300 m) using QGIS 3.16.6 with Grass 7.8.5. All predictor variables, AGB and SOC were transferred to the OSGB 1936 / British National Grid projection geographic information system for analysis.

2.3.1. Topographic Variables

DEM data (EU-DEM v1.1) at a resolution of 25 m was obtained from the Cornipicus Land Portal [26] (2021). It is an upgrade from EU-DEM v1.0, which is generated from SRTM and ASTER GDEM data, through further corrections and improvements. Four DEM derivatives were calculated using QGIS 3.16.6 and SAGA GIS software (Version 2.3.2, SAGA development team, Insitute of Geography, University of Hamburg, Germany), these include elevation, catchment slope (CS), length-slope factor (LSF), and topographic wetness index (TWI).

2.3.2. Inventory Variables

Forest Inventory data are obtained from National Forest Inventory (NFI) [16] developed by the English Forestry Commission. The NFI woodland map provides information on forest and woodland areas with a minimum of 20% canopy cover over 0.5 hectares. The vector data is one hot encoded to represent woodland and nonwoodland classification.

### 2.3.3. Remote Sensing Variables and Processing

The remote sensing data for modeling included S1 and S2 images downloaded from ESA Copernicus Open Access Hub, and LandSat 8 images downloaded from ESGS Earth Explorer. Sentinel 1A data uses SAR (Synthetic Aperture Radar) and records backscatter. This study uses one image using the Interferometric Wide Swath (IW) acquisition mode [23]. The polarization is Vertical Transmit-Vertical Receive Polarisation (VV) and Vertical Transmit-Horizontal Receive Polarisation (VH), which measures volume scattering and rough surface scattering [27]. The image is taken on 5 May 2021, at cycle 230, orbit 52. Sentinel 2A [24] is a wide-swath and multi-spectral satellite. The Multi-spectral Instrument (MSI) samples 13 spectral bands at various resolutions with wavelengths from 442.4 to 2202.4 nanometers [28]. A cloudless Sentinel 2A (Level 1C product) image was captured on 10 October 2018. LandSat 8 carries two sensors, Operational Land Imager (OLI) and the Thermal Infrared Sensor (TIRS). The two sensors provide global coverage at multiple spatial resolutions [25]. The LandSat image was captured on 6 May 2020.

S1 SAR data were pre-processed using the SNAP software: apply orbit file, radiometric calibration, speckle filtering (Lee filter 13 × 13) and terrain correction. To match the resolution of the AGB data, all images were downsampled to 300m using the nearest neighbor algorithm in QGIS 3.16.6. S2 data is processed using the sen2Cor processor which applies atmospheric correction and transform the data product from Level-1C (Top of atmosphere reflectance image) to Level-2A (Bottom of atmosphere reflectance image). Level-L1TP LandSat 8 images were preprocessed through the LandSat Product Generation System (LPGS), which used Ground Control Points (GCP) and DEM to calibrate radiometrically and orthorectify displacements.

The backscatter coefficient from VH and VV polarization in S1 were calculated as environmental variables. Nine bands B2, B3, B4, B5, B6, B7, B8A, B11 and B12 were obtained from S2. Eleven bands L1–L11 were extracted from LandSat 8. Additionally, three spectral indices were calculated from S2 Bands as predictors, these are reported to have a strong correlation with AGB and SOC [14]. These indices are Normalised Difference Vegetation Index (NDVI) [29], Enhanced Vegetation Index (EVI) [29] and Soil Adjusted Total Vegetation Index (SATVI) [30], their formulas are as follows:

$$NDVI = \frac{B8 - B4}{B8 + B4} \tag{1}$$

$$EVI = \frac{2.5 \cdot (B8 - B4)}{B8 + 6 \cdot B4 - 7.5 \cdot B2 + 1} \tag{2}$$

$$SATVI = \frac{2(B11 - B4)}{B11 + B4 + 1} - \frac{1}{2}B12 \tag{3}$$

### 2.4. Modeling Techniques

This paper used three machine-learning techniques to estimate AGB and SOC content. The predictor variables and ground truth variables were first sampled from raw data source raster files and extracted in QGIS 3.16.6. Optimization was performed using grid search in sci-kit learn to tune hyper-parameters. The performance of the models with the best parameters was then evaluated.

### 2.4.1. Random Forest

Random Forest [31] is an ensemble method that predicts through a set of classification and regression trees. These trees are created from a subset of training samples. The in-bag (About two-thirds) samples are used to train trees and the remaining samples are regarded as out-of-the-bag samples and used for evaluation. The error is estimated through an out-of-bag (OOB) error. The prediction of each tree then comprises the final output through voting or averaging.

### 2.4.2. Boosted Regression Tree

The Boosted Regression Tree model combines boosting techniques and a decision tree algorithm for prediction. Boosting reduces overfitting by randomly selecting a subset of training data to fit new tree models. Compared to Random Forest models which use the bagging method, BRTs use the boosting method which weights input data in subsequent trees [32]. Weighting in a way that poorly modeled data in previous trees have a higher probability of selection in the new tree. This improves the accuracy since the model will take into account the error of the previous tree to fit the current tree.

### 2.4.3. XGBoost

Proposed by Chen et al. [33], XGBoost is a very popular ML model upon its success in winning state-of-the-art performance in Kaggle ML competitions. XGBoost is an implementation of gradient-boosted regression trees designed for performance and speed. It uses the second derivative of the objective function to accelerate convergence speed and reduces overfitting by adding a regularization term to the objective function. This results in a highly flexible and scalable model which handles sparse data with high convergence speed.

### 2.5. Statistical Analysis

Statistical analysis is performed to measure collinearity between predictor variables and AGB and SOC. This study used Gini coefficient and Pearson correlation from the SK Learn python package. Variables with high correlation ($r \geq 0.9$) and with high variance inflation factor ($VIF \geq 10$) were removed to form Model E and F. VIF is a ratio between the variance of the model of all variables and the variance of the model of one specific variable. Equations (4) and (5) show the formula for VIF [34] and Pearson Correlation [35] used for our analysis.

$$r_j = \frac{\sum(x_i - \bar{x})(y_i - \bar{y})}{\sqrt{\sum(x_i - \bar{x})^2 \sum(y_i - \bar{y})^2}} \tag{4}$$

$$VIF_j = \frac{1}{1 - r_j^2} \tag{5}$$

The strategy was to eliminate one of the highly collinear variables indicated by VIF scores and Pearson correlation iteratively until all selected variables have a VIF score of less than 10. This paper developed the BRT, RF, and XGB models from the sklearn ensemble methods "Gradient Boosting Regressor", "Random Forest Regressor", and xgboost 1.4.2 from python PyPI repository, respectively.

### 2.6. Model Performance Evaluation

The AGB (Table 2) and SOC (Table 3) content models were built based on three machine learning techniques with different combinations of predictor variables and ground truth variables. A comprehensive list of data sources and their corresponding predictor variables is summarised in Table 4.

**Table 2.** Predicting AGB with different combinations of Sentinel 1, Sentinel 2, LandSat 8, DEM derivatives, forest inventory, AGB and SOC data.

| No. | Model | Data Sources |
|-----|-------|--------------|
| i | Model A | S1, S2 and DEM |
| ii | Model B | L8 and Inventory Data |
| iii | Model C | S1, DEM and Inventory Data |
| iv | Model D | S1, S2, L8, DEM and Inventory Data |
| v | Model F | S1, S2 (Band 4, 8A), NDVI, DEM, L8 (Band 5,6,8,9) and Inventory Data |
| vi | Model H | S1, S2 (Excluding Band 2 and 3), DEM (CS, Elevation), L8 (Band 5–7,10,11), Inventory Data, SOC, SOCD [a] |

[a] Soil Carbon Density (SOCD).

**Table 3.** Predicting SOC with different combinations of Sentinel 1, Sentinel 2, LandSat 8, DEM derivatives, forest inventory, AGB and SOC data.

| No. | Model | Data Sources |
|-----|-------|--------------|
| i | Model A | S1, S2 and DEM |
| ii | Model B | L8 and Inventory Data |
| iii | Model C | S1, DEM and Inventory Data |
| iv | Model D | S1, S2, L8, DEM and Inventory Data |
| v | Model E | S1, S2 (Band 2, 8A), EVI, DEM Derivatives, LandSat 8 (Band 4,5,6,10), Inventory Data |
| vi | Model G | S1, S2, DEM, AGB |

**Table 4.** Data sources and their corresponding predictors.

| Data Source | Environmental Variables |
|-------------|-------------------------|
| Sentinel 1 (S1) | VH, VV |
| Sentinel 2 (S2) | Band 2-7, 8A, 11, 12, EVI, NDVI, SATVI |
| DEM Derivatives | Elevation, CS [a] , LSF [b] , TWI [c] |
| LandSat 8 (L8) | Band 1 - 11 |
| Inventory Data | Woodland category |

[a] Catchment Slope (CS). [b] Length Slope Factor (LSF). [c] Tropical Wetness Index (TWI).

Model A was chosen from the state-of-the-art SOC predictor variables mentioned in Zhou et al. [12], which used S1, S2 and DEM as predictors, and BRT, RF as machine learning algorithms, the paper illustrated the potential of using freely available high-resolution radar (S1) and multispectral satellite data (S2) as input to SOC prediction models. Model B was based on the state-of-the-art AGB predictor variables suggested by Li et al. [15], which used LandSat 8 and Inventory data and discovered XGBoost as one of the more effective machine learning algorithms in predicting AGB, the paper suggested a new method to estimate AGB using remote sensing techniques for subtropical forests in Hunan, China. Model C uses S1, DEM and Inventory data to experiment with the effect of only using a subset of data available including a radar source, one DEM source and Inventory Data, while Model D combined the predictors from Model A and B, and was motivated by the following:

1. Correlation between AGB and SOC suggests a relationship on their corresponding predictor variables [9].
2. While DEM is a useful predictor for SOC, it can also have an effect on AGB as it affects air temperature, moisture and the above-ground growth conditions for trees [36].
3. Above-ground vegetation plays an important role in soil condition and SOC content. Soil organic carbon is found to be richer in forest ecosystems [37], including inventory data in SOC prediction helps better locate forest ecosystems.

Model E and Model F were created after performing statistical analysis from Model D predictor variables on SOC and AGB. On top of statistical analysis, Model G and Model H explored the indirect relationship between AGB and SOC by including them as predictor variables to predict one another target variables. This paper used 5-fold cross-validation to evaluate the performance of the models. Three metrics were used to assess the model's performance. MAE (Mean Absolute Error) [38] and RMSE (Root Mean Squared Error) [38] were used to quantify the difference in error between predictions and ground truth variables, whereas $R^2$ (Coefficient of Determination) [38] was used to quantify how well the model accounts for the variability of input data around its mean. The formulas are demonstrated in Equations (6)–(8). In general, a higher $R^2$ value and lower RMSE/MAE value indicate better estimation performance of the model.

$$RMSE = \sqrt{\frac{1}{n}\sum_{i=1}^{n}(Y_i - X_i)^2} \tag{6}$$

$$MAE = \frac{1}{n}\sum_{i=1}^{n}|Y_i - X_i| \tag{7}$$

$$R^2 = \frac{\sum_{i=1}^{n}(Y_i - \bar{X}_i)^2}{\sum_{i=1}^{n}(X_i - \bar{X}_i)^2} \tag{8}$$

*2.7. Analysis of Results*

The SOC content is converted using a natural logarithm for all prediction models, which reduces the variability of data for more stable training. Through collinearity analysis, there exists high collinearity and correlation in S2 and LandSat 8 variables, all collinear variables with VIF score $\geq$ 10 were removed and reflected in Models E, F, G and H.

## 3. Results

*3.1. Evaluation and Comparison between Models*

The performances for Boosted Regression Tree, Random Forest and XGBoost based on the eight models built are shown in Table 5. Through a comparative analysis of prediction accuracy, it is observed that the different combinations of predictor variables and the choice of machine learning technique significantly affect AGB and SOC prediction performances.

**Table 5.** Prediction accuracy of AGB and SOC with different combinations of predictors. The most accurate results are shown in bold.

| Modeling Technique | Model | AGB | | | SOC | | |
|---|---|---|---|---|---|---|---|
| | | RMSE | MAE | $R^2$ | RMSE | MAE | $R^2$ |
| BRT | Model A | - | - | - | 0.3140 | 0.0968 | 0.7443 |
| | Model B | 173.4170 | 108.7244 | 0.5016 | - | - | - |
| | Model C | 186.2773 | 120.1271 | 0.4180 | 0.3045 | 0.0961 | 0.6887 |
| | Model D | 158.3030 | 100.4707 | 0.5829 | 0.3172 | 0.0973 | 0.7264 |
| | Model E | - | - | - | 0.3792 | 0.1064 | 0.6812 |
| | Model F | **162.8238** | **103.5530** | **0.5898** | - | - | - |
| | Model G | - | - | - | 0.3490 | 0.1038 | 0.6717 |
| | Model H | 163.8379 | 104.0099 | 0.5604 | - | - | - |
| RF | Model A | - | - | - | 0.2944 | 0.0928 | 0.7289 |
| | Model B | 178.5750 | 114.1506 | 0.4958 | - | - | - |
| | Model C | 190.9638 | 124.6872 | 0.4128 | 0.4021 | 0.1141 | 0.5447 |
| | Model D | 161.0494 | 102.9460 | 0.5734 | 0.3398 | 0.0964 | 0.7185 |
| | Model E | - | - | - | 0.3151 | 0.1064 | 0.7295 |
| | Model F | 159.1182 | 101.0034 | 0.5674 | - | - | - |
| | Model G | - | - | - | **0.3075** | **0.0967** | **0.7705** |
| | Model H | **158.6507** | **101.7742** | **0.5925** | - | - | - |
| XGB | Model A | - | - | - | 0.3414 | 0.1129 | 0.6871 |
| | Model B | 168.8985 | 107.0769 | 0.5161 | - | - | - |
| | Model C | 187.8227 | 119.4395 | 0.3965 | 0.3836 | 0.1212 | 0.6100 |
| | Model D | 159.7902 | **100.2105** | **0.5829** | 0.3450 | 0.1131 | 0.7070 |
| | Model E | - | - | - | **0.3239** | **0.1107** | **0.7518** |
| | Model F | 162.5048 | 101.2534 | 0.5604 | - | - | - |
| | Model G | - | - | - | 0.3620 | 0.1158 | 0.6753 |
| | Model H | 160.8522 | 100.2997 | 0.5750 | - | - | - |

### 3.1.1. ML Techniques Evaluation

In AGB predictions, using BRT and RF, Model B ($R^2$ = 0.5016 vs. $R^2$ = 0.4958), Model D ($R^2$ = 0.5829 vs. $R^2$ = 0.5734) and Model F ($R^2$ = 0.5898 vs. $R^2$ = 0.5674) is better predicted by BRT, whereas Model H ($R^2$ = 0.5604 vs. $R^2$ = 0.5925) is better predicted by RF. BRT and XGB have similar performances in Model D ($R^2$ = 0.5829), and XGB performed better in Model B (BRT $R^2$ = 0.5016 vs. XGB $R^2$ = 0.5161) and Model H (BRT $R^2$ = 0.5604 vs. XGB $R^2$ = 0.5750). In SOC predictions, the best results in Model A ($R^2$ = 0.7443) and Model D ($R^2$ = 0.7264) came from BRT, the best result for Model E ($R^2$ = 0.7518) came from XGB and the best result for Model G ($R^2$ = 0.7705) came from RF. Overall, the three machine learning techniques had varying performances with one better than the other in specific models. Figure 2 shows box plots illustrating the % increase in performance across all machine learning techniques for each model compared to the baseline. While different modeling techniques suit different environmental variables in predicting AGB and SOC, we can see a consistent performance increase in AGB performance in Models D, F and H, whereas the improvement for SOC is specific to ML modeling techniques and more research is required to prove consistent improvements.

### 3.1.2. Predictors Evaluation

Throughout the different types of predictors, using S1, S2 and DEM improves AGB prediction by a significant amount. This is reflected when comparing Model B and Model D in all three machine learning techniques: BRT (From $R^2$ = 0.5016 to $R^2$ = 0.5829), RF (From $R^2$ = 0.4958 to $R^2$ = 0.5734), XGB (From $R^2$ = 0.5161 to $R^2$ = 0.5829). For SOC, when comparing Model A and Model D, introducing LandSat 8 and Inventory Data improves performance when modeling with XGB (From $R^2$ = 0.6871 to $R^2$ = 0.7070). However, there

is a slight decrease in performance in BRT (From $R^2 = 0.7443$ to $R^2 = 0.7264$) and RF (From $R^2 = 0.7289$ to $R^2 = 0.7185$) models. Using SOC as a predictor for AGB (Comparing Model D and Model H) improves performance in RF (From $R^2 = 0.5734$ to $R^2 = 0.5925$) and using AGB as a predictor for SOC (Comparing Model D and Model G) significantly improves RF performances (From $R^2 = 0.7295$ to $R^2 = 0.7705$).

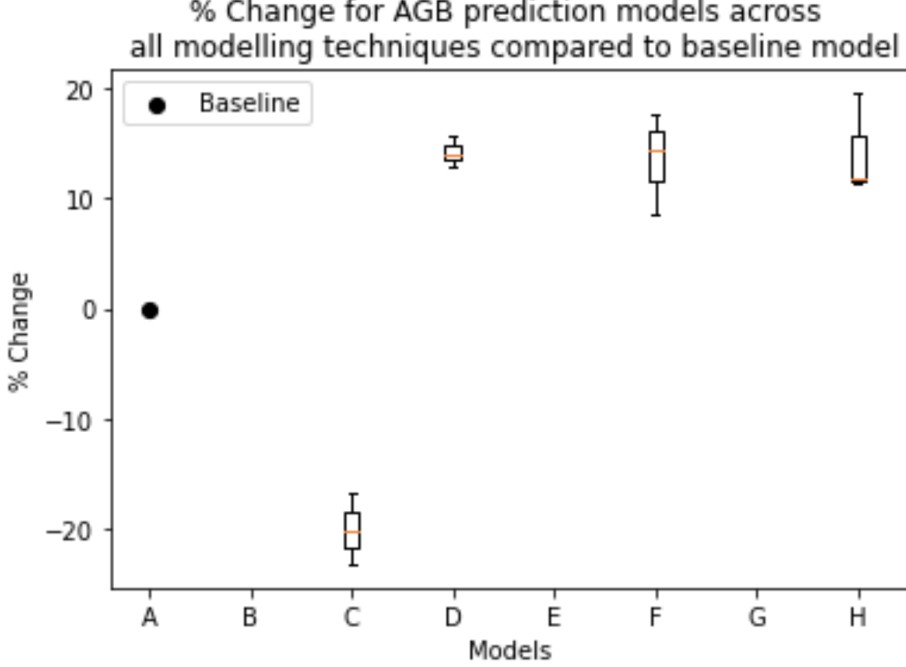

(**a**) AGB Prediction percentage difference compared to baseline

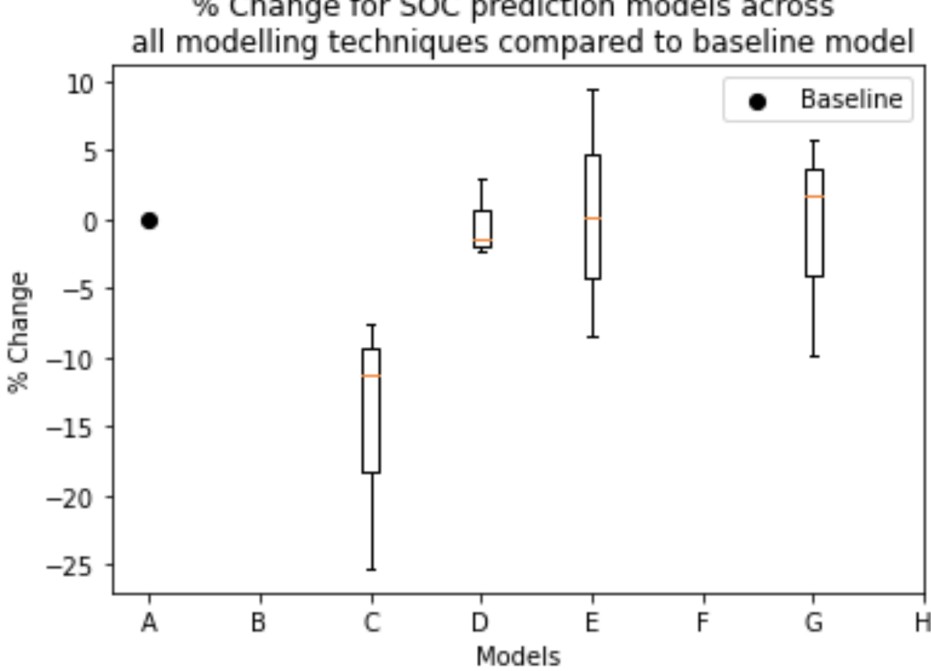

(**b**) SOC Prediction percentage difference compared to baseline

**Figure 2.** Percentage difference of different models compared to baseline models across all machine learning techniques.

Reflected in the results and Figure 2, combining all environmental variables (S1, S2, LandSat 8, DEM, Inventory Data) improves AGB prediction performance by an average of 14.9% (Comparing Model B and Model D, BRT improved by 16.2%; RF improved by 15.7%; XGB improved by 12.9%). This demonstrates that using environmental variables previously known to be good predictors for other target variables can be critical to improving modeling performances. Through applying SOC as a predictor for AGB (Comparing Model A and Model H), the performance in RF improved by 19.5% compared to baseline, while on average an increase of 14.2% (BRT improved by 11.7%; RF improved by 19.5%, XGB improved by 11.4%).

### 3.2. Feature Importance of Predictors

For AGB and SOC mapping with Model D, Model H and Model G, the percentage relative importance of predictor variables are shown in Figures 3 and 4. Overall, BRT and XGB models depend heavily on one or a few predictors while RF models allow importance spreading across a wider range of variables. The AGB model predictions are heavily influenced by Inventory data, which is to be expected given that AGB is predominantly found in woodland areas. Notable is the fact that both Sentinel 2 and DEM derivatives contribute to the predictive power of AGB BRT Model D. This evidence substantiates our claim that combining SOC and AGB predictors improve AGB model estimation. In SOC models, we can see that Band 8A has the greatest impact on prediction, while Sentinel 1 data and digital elevation also play a role. In Model G, although not the most important factor, AGB still plays a role in SOC estimation and its importance is comparable to that of Sentinel 2 Bands (2–5). This explains the slight improvement in SOC estimation from RF Model D ($R^2$ = 0.7185, MAE = 0.0964, RMSE = 0.3398) to RF Model G ($R^2$ = 0.7705, MAE = 0.0967, RMSE = 0.3075), although AGB has some influence, it is not the most important variables in predicting SOC. The model prediction is still dominated by Sentinel 1 data and Band 8A from Sentinel 2. For Model H, inventory data still has a very large influence in the model as expected, followed by Sentinel 2 and Landsat 8 data. It is interesting to see that Soil Carbon Density is now more important than Digital Elevation and Sentinel 1 data, verifying the positive correlation between AGB and SOC discovered in previous literature [7,9]. Despite the correlation discovered, Rasel [9] only experimented the possible effect of AGB on SOC Estimation. We have now proved the other way around as well, using SOC and SOC Density improves AGB Estimation performance.

### 3.3. Spatial Characteristics of AGB and SOC Maps

Carbon maps for AGB (Figure 5) and SOC (Figure 6) are obtained from Model H and Model G predictions, respectively. The total carbon map in Figure 7 is generated by adding carbon predictions in both maps together. The total carbon error map in Figure 8 is created by the absolute difference between the predictions and ground truth carbon content. This can be compared against the AGB ground truth map in Figure 9, SOC ground truth map in Figure 10 and the Total Carbon ground truth map in Figure 11.

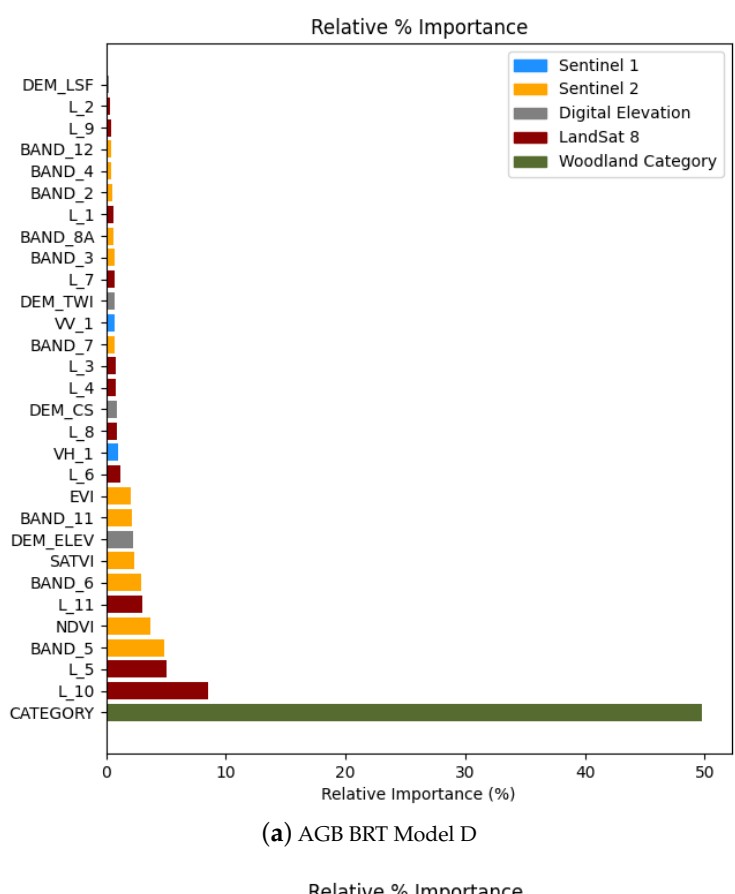

(**a**) AGB BRT Model D

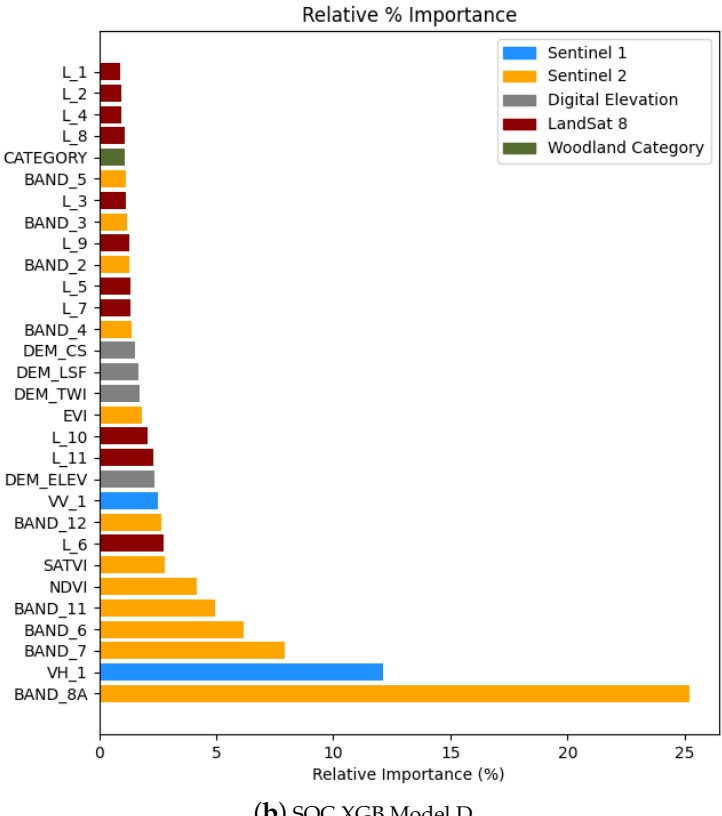

(**b**) SOC XGB Model D

**Figure 3.** Feature Importance in Models (Model D).

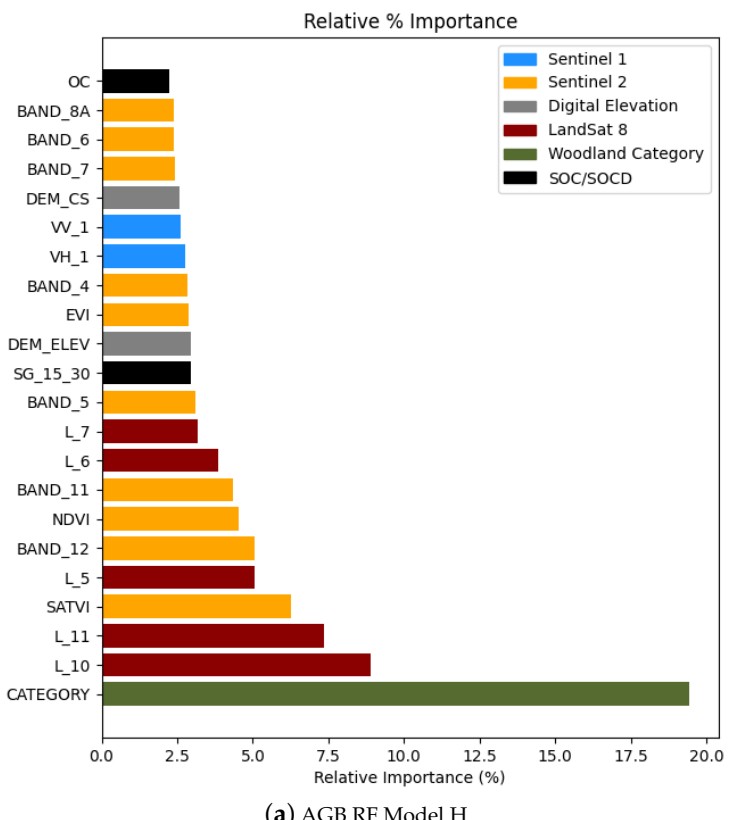

(**a**) AGB RF Model H

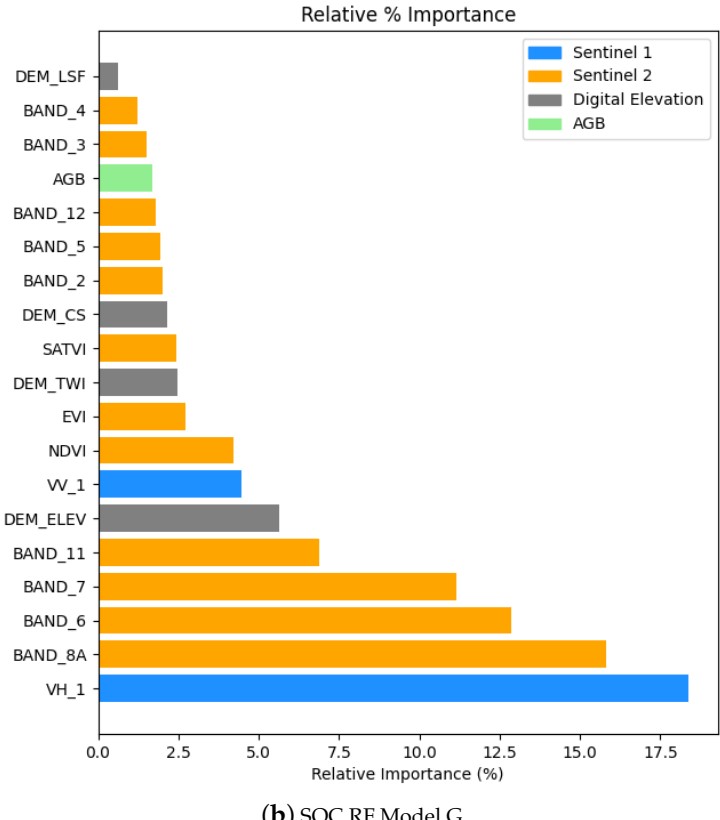

(**b**) SOC RF Model G

**Figure 4.** Feature Importance in Models (Model H and G).

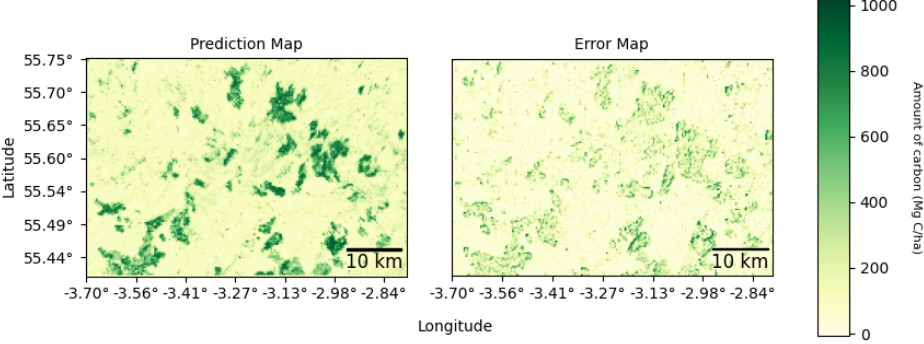

**Figure 5.** AGB Carbon Prediction and Error Maps.

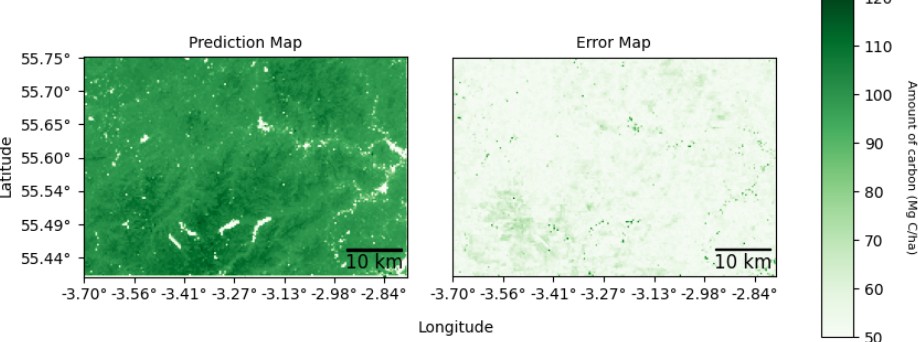

**Figure 6.** SOC Carbon Prediction and Error Maps.

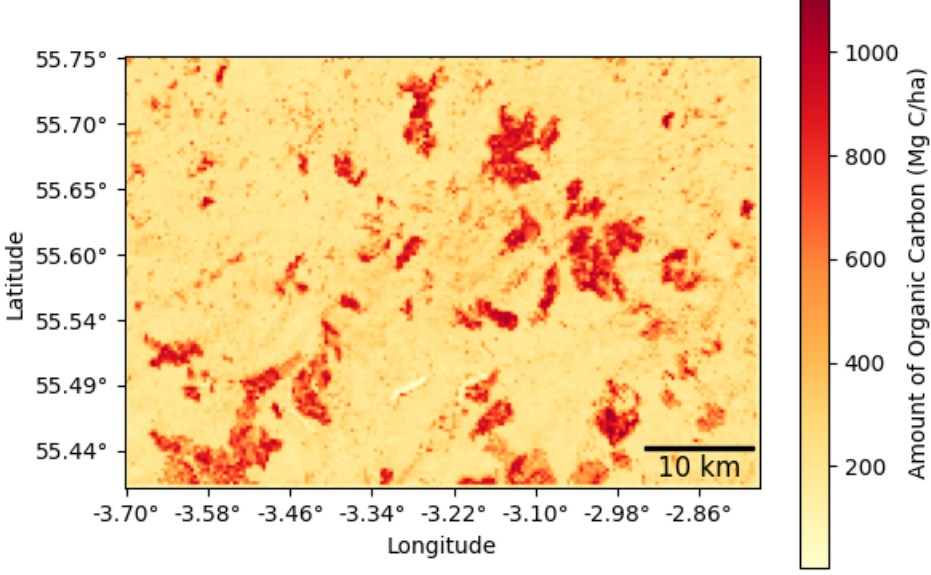

**Figure 7.** Total Carbon Prediction Map (AGB and SOC).

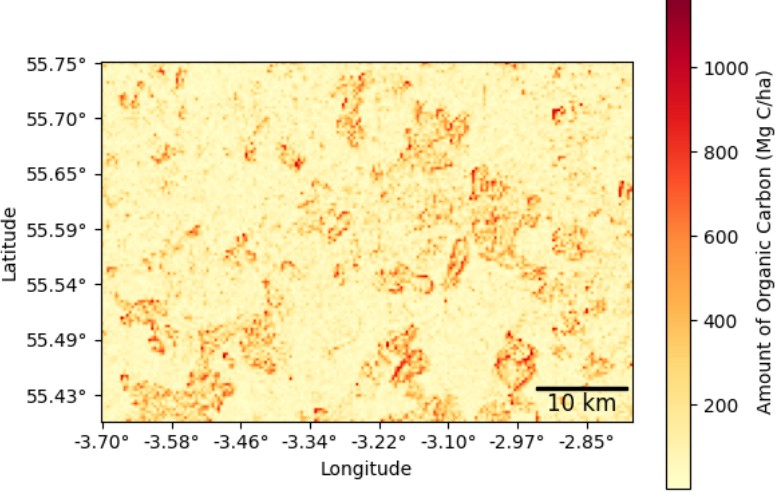

**Figure 8.** Total Carbon Error Map (AGB and SOC).

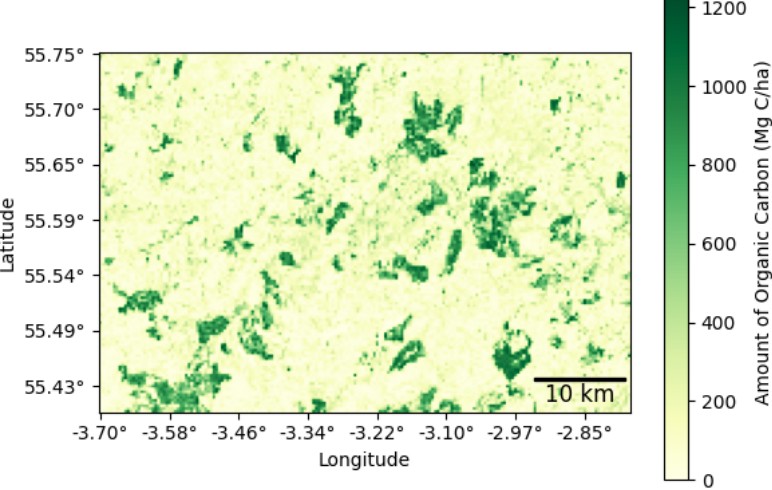

**Figure 9.** AGB Ground Truth Carbon Map.

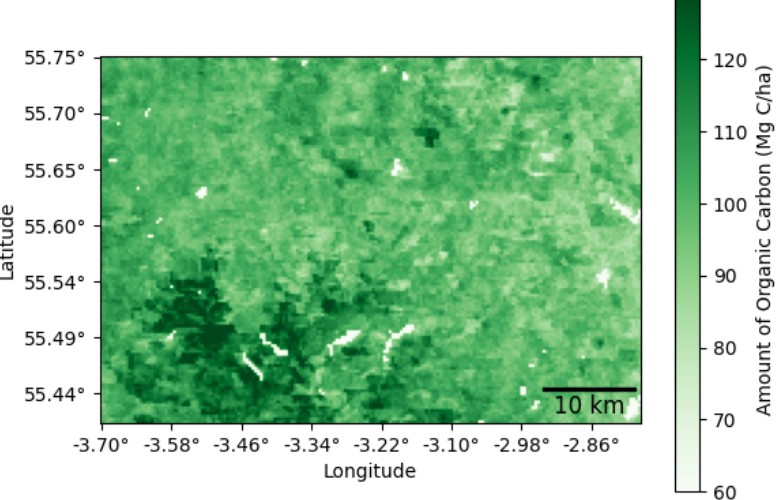

**Figure 10.** SOC Ground Truth Carbon Map.

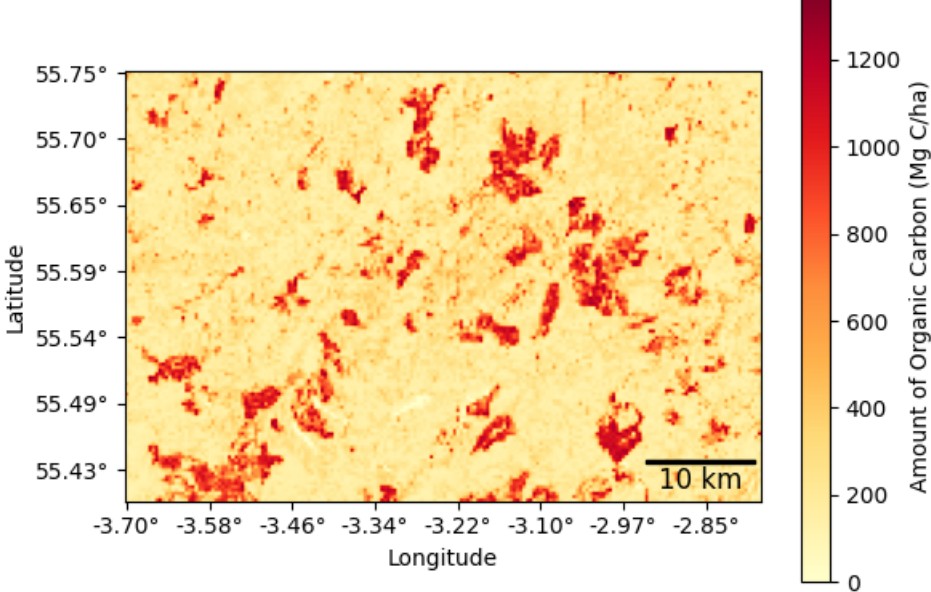

**Figure 11.** Total Carbon Ground Truth Carbon Map.

## 4. Discussion

*4.1. Performance of Prediction Models Using Sentinel 1, Sentinel 2, LandSat 8, DEM and Forest Inventory Data*

**Slight difference in SOC union models compared to baseline models:** For the SOC models, Models A (baseline) and D (union model) perform similarly across all three modeling techniques, BRT shows a difference of ($R^2$ = 0.0179, MAE = 0.0005, RMSE = 0.0032), RF shows a difference of ($R^2$ = 0.0104, MAE = 0.0036, RMSE = 0.0454), XGB shows a difference of ($R^2$ = 0.0199, MAE = 0.0002, RMSE = 0.0036). The slight change in performance suggests predictors used in AGB estimation are not very useful to predict SOC. This can be explained by the fact that forest inventory data only identify forest areas [16] but did not take into account the fact that soil organic carbon is also abundant in other land covers such as agricultural land.

**VIF collinear variable removal improves performance:** Lombardo et al. [39], suggested removing one of two or more collinear variables iteratively to avoid multicollinearity. Following this method, we created Model E (predicting SOC) which improved the XGB technique for SOC from the ($R^2$ = 0.6871, MAE = 0.1129, RMSE = 0.3414) baseline to ($R^2$ = 0.7518, MAE = 0.1107, RMSE = 0.3239) and Model F (predicting AGB) which improved the BRT technique from The BRT technique for AGB also improved from ($R^2$ = 0.5829, MAE = 100.4707, RMSE = 158.3030) to ($R^2$ = 0.5898, MAE = 103.5530, RMSE = 162.8238) On the contrary, if we remove all collinear variables (Model C) at once, the predictive power decreases for all modeling techniques. This is because removing multiple collinear variables simultaneously has the unintended consequence of removing information that is not highly collinear with the remaining variables. It is important to note that, although not experimented with in this paper, the feature selection process can also leverage domain knowledge from SOC or AGB experts which can complement these ML techniques.

**Using Digital Elevation Data in AGB estimation models significantly improves performance:** This is one of the major findings as there is a significant improvement in AGB model performances after including variables previously used for SOC model predictions. Zhou et al. [12] used Sentinel 1, Sentinel 2 and DEM data to predict SOC while Li et al. [15] used Sentinel 1 and LandSat 8 to predict AGB. Using predictors previously used in SOC prediction improves the AGB model by a significant 14.9% across all ML techniques (Shown in Table 5 and illustrated in Figure 2). This indicates that Sentinel 2 and DEM data contain useful information for predicting AGB. There is no prior attempt in any literature to use

Digital Elevation to estimate AGB. It demonstrates that factors associated with SOC may have an effect on predicting AGB.

*4.2. Spatial Characteristics of Prediction Maps*

From the total carbon and error maps, most prediction errors come from above-ground biomass concentrated regions, while we are very successful in predicting the locations of high carbon content regions, the estimation in these regions still requires more attention. There are two ways to mitigate this problem and improve our carbon map performance.

1.  **Higher resolution study at specific regions of interest:** We encounter noisy data when attempting to map the entire region which consists of a mix of land use [16]. If we can perform segmentation [40] and target regions with high carbon content, then we can eliminate unnecessary noise and obtain better results.

2.  **Remove area that is impossible to have above-ground biomass:** While this might not be the case for SOC, it is possible to identify areas with no above-ground biomass and eliminate those regions from our study. For instance, it can be clearly identified that roads and urban areas have no above ground biomass value [16]. We can set the AGB values and the predictor values for those regions to 0. This is another way to remove noise such that our model can focus on predicting the highly carbon-concentrated regions.

*4.3. Novel Discoveries*

Table 5 extracts the results for our joint study models. The random forest model beats the state-of-the-art result by 19.5% in AGB estimation and by 14.2% on average across all ML techniques (Comparison between RF Model B and Model H). This is consistent with the observation that there is a direct positive relationship between AGB and SOC [9]. We were able to verify the correlation between AGB and SOC despite our study area consisting of a mix of forest and agricultural land. This is expected to be more prominent if we restrict our study area to only forest areas [7]. We have demonstrated that using SOC/SOCD to predict AGB improves model results. Thus, a joint study between AGB and SOC is a crucial direction for future research in the domain of total carbon estimation.

4.3.1. Digital Elevation as Predictor Improves AGB Estimation

Through the experiment of mixing AGB and SOC predictors, we observed a significant increase in performance in AGB estimation through the use of Digital Elevation Map as a predictor. With an average increase of 13.53% across all three ML techniques, we discovered a way to leverage the relationship between AGB and SOC to improve machine learning model results. This insight helps future studies in the total carbon domain to identify the most important predictors for carbon estimation models.

4.3.2. SOC and SOC Density Are Good Predictors for AGB Models

We experimented using SOC and SOC Density as predictors for AGB estimation models, the best-performing machine learning technique increases performance from $R^2 = 0.5829$ in RF Model D to $R^2 = 0.7705$ in RF Model H.

4.3.3. Using AGB as a Predictor for SOC Improves Performance for Certain ML Techniques

We discovered the indirect relationship between AGB and SOC, upon using AGB as a predictor variable, we improved model performance from $R^2 = 0.7185$ in RF Model D to $R^2 = 0.7705$ in RF Model G. However, when taking into account other ML techniques, there is no improvement on average, the improvement is therefore ML technique specific and more research is required. On the other hand, upon performing feature importance analysis, we discovered that AGB has a certain importance in the SOC estimation model.

*4.4. Insights*

This study has made a significant contribution to the field of AGB prediction by exploring the integration of remote sensing (RS) and digital elevation model (DEM) data. A key finding from this research is that the incorporation of DEM data can substantially enhance AGB prediction accuracy when used in combination with RS data. While RS data provides critical information about landforms and vegetation, DEM data add valuable insight into land morphology [41], including topographic depression and flow direction. These features are crucial for determining the topographical wetness index (TWI) and identifying drainage areas in landforms [42], which significantly contribute to AGB prediction accuracy.

The use of RS data alone has been limited by several factors, including low resolution, availability and cost of data and limited feature extraction. However, the incorporation of DEM data into AGB prediction models has shown that this can overcome some of these limitations and enable a more comprehensive understanding of AGB. The results demonstrate that DEM data should not only be used for predicting soil organic carbon (SOC) but also integrated into AGB prediction models to ensure accurate predictions.

One of the significant benefits of combining RS and DEM data is that it provides a more holistic approach to AGB prediction. This combination allows for the identification of vegetation areas that may have been overlooked due to resolution limitations or feature extraction constraints. Furthermore, the DEM data provides a wealth of information on land morphology, which is critical for determining the TWI and identifying drainage areas [43]. The inclusion of this information into AGB prediction models results in more accurate and reliable predictions.

The study highlights the need for future research to consider the integration of DEM data to improve the accuracy of AGB prediction models further. As demonstrated in this study, the use of DEM data in combination with RS data can overcome the limitations of RS data alone and provide a more comprehensive understanding of AGB. By addressing these limitations and employing a more holistic approach to AGB prediction, researchers can improve the accuracy and reliability of AGB prediction models, which have important implications for ecosystem management and climate change mitigation.

## 5. Conclusions

In this work, we proposed a general methodology to estimate the total carbon content in an AFOLU area of Scotland. There are two novel experiments conducted that contribute to the remote sensing carbon estimation domain: **(i) Create a union predictor model that consists of predictors from the SOC and AGB carbon estimation domain. (ii) Explore the indirect relationship between SOC and AGB and improved carbon estimation performance through the use of target variables as predictors.** The experimentation results suggest that a joint study of AGB and SOC is important for carbon estimation as biomass and soil continuously exchange carbon in terrestrial ecosystems. Through feature engineering and the two novel experiments we conducted, we improved the state-of-the-art AGB estimation by 14.2% on average across all ML modeling techniques discussed (When comparing the $R^2$ value across all three modeling techniques in Model B and Model H, the is an increase from 0.5016 to 0.5604 for BRT, 0.4958 to 0.5925 for RF and 0.5161 to 0.5750 for XGB).

**Author Contributions:** Conceptualization, C.K.C., C.A.G., A.K. and P.M.B.-V.; methodology, C.K.C.; software, C.K.C., C.A.G., A.K.; validation, C.K.C.; formal analysis, C.K.C.; investigation, C.K.C.; resources, C.K.C., C.A.G., A.K. and P.M.B.-V.; data curation, C.K.C., C.A.G., A.K.; writing—original draft preparation, C.K.C.; writing—review and editing, C.K.C.; visualization, C.K.C.; supervision, P.M.B.-V.; project administration, C.K.C.; funding acquisition, P.M.B.-V. All authors have read and agreed to the published version of the manuscript.

**Funding:** This research was funded by the Royal Society grant number ERR\19\104 and we thank you J. McCann for their support.

**Data Availability Statement:** https://github.com/TerrenceCKCHAN/Carbon-Trading-Verfication, accessed on 27 September 2021.

**Acknowledgments:** We thank you Thomas Lancaster for his advice on project direction and possible improvements. Harry Grocott and Rob Godfrey from Treeconomy for their time discussing useful data sources and sharing their expertise in remote sensing. Engineering Change for their input with innovative ideas and ideology of an application to showcase carbon maps.

**Conflicts of Interest:** The authors declare no conflict of interest.

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
