# Peer review of "Satellite-Based Carbon Estimation in Scotland: AGB and SOC"

_land, doi:10.3390/land12040818_

Round 1
Reviewer 1 Report
In this work, a general methodology was proposed to estimate total carbon content in the AFOLU region of Scotland. This paper creates a joint predictor model consisting of predictors from the SOC and AGB carbon estimation domain, and explores the indirect relationship between SOC and AGB and improved carbon performance assessment by using the target variable as predictors. The result shows that joint study of AGB and SOC is important for carbon estimation as biomass and soil continuously exchange carbon in terrestrial ecosystems. This paper is trying to combinate extra variables to improve SOC and AGB predication accuracy. This paper can be accepted after handled the following issues.
1. The Abstract and Introduction sections are too detailed described by authors, they need concisely described.
2. In the Introduction section, the authors introduced every issue with subtitle, this is more like a thesis, especially the authors give the concepts of several words like AGB and SOC, it is needn’t.
3. The model A and B were referred from “The state of the art” of Zhou et al., and Li et al., some detailed information need to be described.
4. Some detailed information need to be expressed for the study area such as climate, topography, landforms.
5. The date of the data in this paper need to be indicated.
6. Some meanings of the characterizes in the formulas need to be indicated clearly.
7. Model C need to be explained specifically.
8. In the Results section, 3.1 are the explanation of how to deal with the model results that it should be moved to Materials and methods section.
9. Repetition was detected in lines 311-318 with M&M 2.6.
10. A very long and detailed results was expressed in 3.2, it can be separated into several subtitles and concised.
11. As the important results of this paper, Figure 4 to Figure 7 need to be presented a bit detailed.
12. Not a comprehension discussion was expressed in the paper, most of the contents are repletion of the Results. Why and how the predication accuracy was improved by joint the RS and DEM data.
13. Table 8 is part content of Table 7, it can be deleted.
Author Response
See attached Cover letter

Reviewer 2 Report
Land
Manuscript ID: 2263581
Joint Study of Above Ground Biomass and Soil Organic Carbon for Total Carbon Estimation using Satellite Imagery in Scotland
by
Chan, Gomez, Kothikar, Baiz
REFEREE’S COMMENTS
This is an interesting paper on a subject that should be of great interest to many readers. For location of comments, see the belows.
- Title:
- It should be more concise/compact.
- Abstract:
- Introductory level sentences should be removed.
- The authors should give their basic findings in a quantitative manner.
- Introduction:
- Figure 1 should be given in a more clear way for potential international readers.
- See the paper on https://doi.org/10.5194/nhess-10-89-2010, 2010.
- Literature review on the "machine learning" should be given as well.
- Materials and methods:
- line 206-211; refer relevant papers already available in the literature.
- Refer relevant papers for the equations derived in the text.
- Results:
- Use capital letter (F) in the word "figure" through the text.
- Formatting of the Figures given in the submission should be consistent with each other.
- Figure 3 is not seen clearly.
- Discussion:
- line 422-437; refer relevant papers already available in the literature.
- Conclusions:
- The authors should quantify their findings in this section.
- In General:
- Check out the details of the references cited.
Best regards,
Author Response
See attached cover letter
